# Energy Transition in France

Badr Eddine Lebrouhi [1] , Eric Schall [1], Bilal Lamrani [2], Yassine Chaibi [1,*] and Tarik Kousksou [1,*]

1   E2S UPPA, SIAME, Université de Pau et des Pays de l'Adour, 64000 Pau, France;
b.lebrouhi@etud.univ-pau.fr (B.E.L.); eric.schall@univ-pau.fr (E.S.)
2   Faculté des Sciences, Laboratoire MANAPSE, Université Mohammed V de Rabat, B.P. 1014 RP,
Rabat 10000, Morocco; b.lamrani@um5r.ac.ma
*     Correspondence: chaibi.yassine@gmail.com (Y.C.); tarik.kousksou@univ-pau.fr (T.K.)

**Abstract:** To address the climate emergency, France is committed to achieving carbon neutrality by 2050. It plans to significantly increase the contribution of renewable energy in its energy mix. The share of renewable energy in its electricity production, which amounts to 25.5% in 2020, should reach at least 40% in 2030. This growth poses several new challenges that require policy makers and regulators to act on the technological changes and expanding need for flexibility in power systems. This document presents the main strategies and projects developed in France as well as various recommendations to accompany and support its energy transition policy.

**Keywords:** renewable energy; hydrogen energy; energy transition; energy storage; energy policy

## 1. Introduction

Energy security has been a crucial issue in the energy paradigm for the last 20 years [1,2]. In recent years, our energy system has been significantly transformed [3]. Among the many developments, we can note the substantial fluctuations in crude oil prices, with the record peaks reaching in July 2008 and the unexpected collapse of the barrel in 2014; the rapid development of shale gas in the United States; the European energy market liberalization; as well as the Fukushima accident and the impacts of this tragedy on specific national energy strategies [4]. In 2009, for the first time since 1945, a net reduction in global electricity consumption was registered. In 2020, the appearance of the COVID-19 pandemic constituted a massive shock, which plunged the world economy into a more severe recession than that recorded during the 2009 crisis [5]. The recent conflict in Ukraine has caused European natural gas prices to reach a new historical record. Supply from Russia could be seriously affected by possible economic sanctions directly targeting the Russian energy sector.

Despite the worldwide COVID-19 lockdowns leading to a decrease in fossil fuel use; 2020 matched 2016 for the highest global temperatures ever recorded. Copernicus experts determined that the last six years were the warmest six on record [6]. Last year, the average surface temperature across the planet was around 1.25 °C higher than the 1850–1900 pre-industrial period. The year 2020 was also 0.6 °C warmer than the average from 1981–2010.

In this context, during the World Climate Conference COP26 recently organized in Glasgow, experts responded to the worrying data by calling for urgent action to slow down global warming and keep the world within 1.5 °C of warming. The European Commission's Matthias Petschke said: "The extraordinary climate events of 2020 and the data from the Copernicus Climate Change Service show us that we have no time to lose". The latest IPCC (Intergovernmental Panel on Climate Change) report clearly revealed the magnitude and rapidity of the observed changes, which, in some cases, surpass previous forecasts. Limiting global warming to 1.5 °C requires great and urgent efforts to modify public policies and our lifestyles.

The heavy use of fossil fuels has already had irreparable effects, and humanity is placing itself in an unprecedented danger zone. How can we reconcile the increase in energy demands with the urgent need to reduce our carbon footprint? Only one form of energy seems to be viable and responsible in the future, for which the entire world has to participate in its development: decarbonized electricity [7].

Several governments worldwide have implemented transition policies aimed at drastically reducing $CO_2$ emissions related to energy production and consumption [8–16]. All of this is taking place in a context marked by a general trend towards the decentralization of the means of production and, for the economy as a whole, by an increasingly strong emphasis on the digitalization of processes [17].

According to the "Announced Policies" scenario elaborated by the World Energy Outlook (WEO), more than half of the additional electricity production would come from wind and photovoltaic power by 2040, and practically 100% under the "sustainable development" scenario presented by the International Energy Agency (IEA) [18]. However, the IEA notes that whatever the trajectory taken by the energy system, the world continues to strongly depend on oil supplies from the Middle East [19]. Recently, the IEA developed this scenario further in a special report proposing a roadmap to achieve global CO2 neutrality from the energy system by 2050. Published in 2021, this report entitled "Net Zero by 2050" marks a rupture in the IEA's doctrine, which, until then, had never accepted to consider a decrease in energy consumption at the global level in its prospective studies. In this study, the world's wealth doubles by 2050, the population increases by two billion, but energy demand falls by 8%. In this decarbonized scenario, electricity covers half of humanity's energy needs (compared to just under 20% today). Nuclear power will play a marginal role, only 10% of the electrical mix in 2050. Conversely, 90% of electricity production will be provided by renewable means, including 70% for solar and wind power.

In France, the Energy Transition and Green Growth Law fixed a target of 32% renewable energy in final energy consumption by 2030, and 40% in final electricity consumption [20,21]. Such growth presents many new challenges that require decision makers and regulators to quickly act to respond to the rhythm of technological evolutions and the growing flexibility needs of electrical systems [22–25].

Energy transition and carbon neutrality have received a lot of attention in recent years, and an increasing number of articles have recently been published [26–31]. A number of them highlight the policies on this crucial topic in different countries (namely, Italy [32,33]; the United kingdom [34]; Spain [35]; Morocco [10]; Germany; China [7,15,36,37]; Portugal [38]; Canada [39]; the United States [14]; India [40]; South Korea [41]; Botswana [42]; South Africa [43]; Mexico [8]; Switzerland [11]; and Russia [12]), regions (i.e., Europe [44], Africa [45], South America [17], and South Asia [46,47]), and a select few compare policies across countries or regions (namely, France and Sweden [48]; Spain and Portugal [49]; Germany and China [50]; China and Africa [51]; two Austrian regions [13]; Global North and Global South [16]; Southeastern Europe and the Commonwealth of Independent States (CIS) [9]; lower- and middle-income countries [52]; developing countries [53]; Australia; Brazil; Canada; China; Europe; India; Indonesia; Japan; The Republic of Korea; Russia; and the United States [54]).

As far as France is concerned, previous reviews have focused on energy law and policy [55,56], needed raw materials [57], or a comparison of the energy transition in France to other countries [42]. The articles previously cited do not address, in detail or in depth, the various recent policies and initiatives implemented in France in all sectors. The objective of this article is to present, in depth, the main reforms carried out in Europe, as well as the different solutions and measures developed at the level of each sector in France to support its energy transition policy.

## 2. Energy Sector Reforms in Europe

The basis of European policy is the definition of directives that include measures that each Member State must transpose into its legislation, with a certain level of subsidiarity

that allows each Member State's particularities to be considered and authorizes certain flexibility. European norms and regulations are added to these directives. The energy sector and the electricity market in Europe have undergone several successive reforms over several years [55–58]. In 1997, a liberalization process of the electricity market was initiated (Directive 96/92/E.C.). In 2002, the European Union Council signed an agreement that envisages the liberalization of such markets for all customers by 1 July 2007, at the latest. In 2008, the European Union set its Energy and Climate Package with three objectives called 20–20–20 to be achieved by 2020:

- A 20% reduction in greenhouse gas emissions compared to 1990;
- A total of 20% of the total final energy consumption produced from renewable energy sources;
- A total of 20% energy efficiency compared to the reference scenario.

While the achievement of the two first objectives is mandatory, the third remains an indicative and non-constraining target. In 2012, the E.U. adopted a directive on energy efficiency, which imposes on E.U. states the obligation to set national energy efficiency targets and impose a renovation constraint of 3% of the floor space of buildings owned and occupied by the central government, an obligation to define a roadmap for the renovation of tertiary and residential buildings and the liberalization of energy markets to demand management programs. In 2014, the European Council set the following objectives for 2030:

- A 40% reduction of greenhouse gas emissions compared to 1990;
- A total of 27% of the total final energy consumption produced from renewable energy sources;
- A total of 27% energy efficiency compared to the reference scenario.

In 2015, the European Commission launched the Energy Union project to reinforce electricity interconnections between the Member States by linking all of the national transmission systems. This major project aims to promote energy exchanges between countries in order to reduce Europe's dependence on gas imports and work towards a better integration of intermittent renewable energies. One of the consequences of the development of market coupling solutions, in conjunction with the development of interconnections, is to reduce the differences in the prices within interconnected regions, even though sometimes significant differences may remain between regions. In December 2015, the Paris World Climate Conference COP21 (21st Conference of the Parties) united 195 countries. A commitment to limit the rise in temperatures below 2 °C was signed.

The "Clean Energy for all Europeans" project proposed by the European Commission at the end of 2016 represented an essential step in constructing the Energy Union [59]. This project aims to accelerate the integration of electricity markets in Europe, to continue the development of renewable energies, and to promote energy efficiency, while placing the European consumer at the center of the energy transition process. Among the quantitative objectives to be achieved within the framework of this legislative package by 2030 are the following goals: to reduce greenhouse gas emissions by 40%, increase the share of renewable energies to 32%, improve energy efficiency by 32.5%, and increase electrical interconnection capacities by 10 to 15%.

Under the impetus of the new President of the Commission, Ursula von der Leyen, Europe is also accelerating the low-carbon transition. During her speech in Parliament at the end of 2019, President von der Leyen announced her Green Deal project, aiming to achieve carbon neutrality by 2050. This project has following four main components [60]:

- To increase financing, both public and private, to accelerate the low-carbon transition. To establish a USD 100 billion public fund to finance the industrial conversion, particularly in the coalfields.
- An upward revision of the greenhouse gas emission reduction targets for 2030 to at least 50% and, if possible, 55%. The European objectives, duly deposited in the registers of the Paris Agreement, are to achieve a 40% reduction in 2030 relative to

1990, a 32% increase in the share of renewable energies, and a 32.5% improvement in energy efficiency.

- The reinforcement of carbon pricing via the $CO_2$ quota trading system. To achieve the new 2030 objectives, the Commission plans to mobilize the carbon market in two ways: a further reduction in the emission cap, which will undoubtedly drive up the price of the allocation, and an enlargement of the perimeter covered by the system, which could, if Parliament agrees, cover all energy-related $CO_2$ emissions in the long term.
- The establishment of a frontier adjustment mechanism. As a result, it is becoming problematic to reconcile a high-climate ambition with free-trade rules. The major challenge for Europe would be to subordinate the traditional rules of free trade to the priorities of its climate policy.

Recently, the European Union has implemented a policy (through its Green Deal program) to produce green hydrogen in two stages [61–63]:

- By 2024, at least 6 G.W. of electrolyzers in the E.U., producing 1 $MtH_2$/year.
- By 2030, at least 40 G.W. of electrolyzers, producing 10 $MtH_2$/year.

To achieve these objectives, the E.U. needs to heavily invest in wind and solar power, as well as import green hydrogen from other countries. For this reason, the European Union has implemented the green taxonomy concept to build a sustainable finance ecosystem. One of the main objectives of this concept is to orient private investments towards sustainable activities to enable the European Union to reach climate neutrality by 2050. According to the European taxonomy, gas and nuclear power are eligible for green finance to support the energy transition in Europe. However, certain limits and conditions have been imposed in order to include nuclear energy as a part of sustainable energy. New nuclear power plant construction must have a building permit before 2045. Projects to extend the life of existing plants will have to be authorized before 2040. Guarantees for waste treatment and decommissioning of nuclear facilities at the end of their life will also be mandatory.

In Europe, over the last few years, renewable electricity production has increased (wind (+54%) and photovoltaic (+10%)), thanks to the improvement of load factors and the increase in installed production capacities [64]. This trend reflects the voluntary policies that the Member States has implemented to meet the 2030 targets of the energy-climate package and the continuous reduction in production costs in the renewable energy sector. This has contributed to a further decline in the output of coal-fired power plants and lower electricity prices on wholesale electricity markets.

## 3. The Energy Context in France

### 3.1. The Energy Mix in France

France's energy mix is one of the most decarbonized globally, thanks to nuclear energy, which represents the main source of electricity domestically produced (more than 70% of the total production in 2019) [48,65,66]. As a primary means of production, nuclear power produces a relatively stable amount of electricity throughout the year. When examining the age of France's nuclear power plants, we can observe that a substantial part of them were commissioned between the end of the 1970s and the beginning of the 1980s, so they are between 33 and 40 years old [67]. The average lifetime of a nuclear power plant is of the order of 40 to 60 years; therefore, in the coming years, a significant number of plants could be gradually closed down [68] (see Table 1).

**Table 1.** The nuclear power plants in operation.

| | |
|---|---|
| Number of reactors | 56 |
| Park power | 61.4 GW |
| Average age of the reactors | 36.3 years |
| Reactors reaching 40 years old | 9 |

To support the energy transition process, France opted to reduce the share of nuclear power in the electricity mix [69]. Therefore, the law adopted in 2015 consisted of limiting the installed nuclear capacity (at its current level of 63.2 GW) and setting a target of 50% nuclear power in electricity production by 2025 (compared to about 75%, currently). In its 2016 annual report [70], the French National Audit Office estimated that achieving this objective would require the closure of 17 to 20 reactors, assuming a stable future electricity demand, equivalent to one-third of the present installed capacity. The rate of closure of nuclear power plants in France was one of the central debates held in preparation for the new France's multi-year energy plan (known by its French acronym PPE) for the period of 2018–2023 [20]. The PPE project, published by the government at the end of January 2019, calls for the closure of 4 to 6 reactors (including the two reactors at Fessenheim) by 2028, and the achievement of the 50% target by 2035 with the closure of 14 reactors. This new target of 50% has been ratified by the Energy and Climate Law adopted in September 2019 and promulgated in November [71]. The extension of the operating life of 32 reactors with a capacity of 900 MW beyond 40 years was recently approved by the French Nuclear Safety Authority (ASN), which published a recommendation to this effect in February 2021. The PPE also mentioned the possibility of renovating part of the existing nuclear power plants by building 14 new EPRs (Evolutionary Pressurized Reactors).

Two scenarios were recently expressed in France to outline the 2050 energy mix:

- The first prospective scenario was presented by the negaWatt association [72]. It formulated proposals to achieve carbon neutrality in 2050 based on three pillars, sobriety, efficiency, and renewables, while reducing the extraction of raw materials. In this scenario, none of the 56 reactors in operation extend beyond 50 years. In 2045, no nuclear power will exist. According to this scenario, 100% renewable generation can be achieved without destabilizing the system. This can be achieved through the complementarity of sources, the flexibility of consumption, and the development of energy storage systems. Additionally, this will be achieved while preserving the raw materials in France, thanks to the sobriety and efficiency actions carried out in all of the sectors, but also to the substitution of non-renewable materials by bio-based materials and the increase in recycling rates. Only lithium consumption will increase to support the development of electric vehicles and electricity storage.

- The second scenario was prepared by RTE (Réseau de Transport de l'Electricité) and is called "Energy Futures 2050" [73]. Six scenarios were presented in Energy Futures 2050. Three scenarios prioritize renewables and the other three nuclear power. The different scenarios are presented to meet the demand by 2050, estimated from 555 TWh, if society shifts towards more sober lifestyles, to more than 750 TWh, if it remains very energy intensive. The reference of the demand used in Energy Futures 2050 is 645 TWh.

Despite its significant potential—Europe's second largest wind resource, second largest exclusive maritime zone, fifth largest source of sunshine, and abundant forest and hydropower resources—the share of renewables for the final energy consumption in France was only 16.5% in 2018 [20,21]. However, this share should be approximately 20.5% to comply with the 2020 objectives (see Figures 1–3) [73]. The target of 23% of RE for the final energy consumption in 2020 suffered from the slow progress of offshore wind energy. Not all sectors have the same tendencies [74]. Heat pumps and geothermal energy for renewable heat and cold are beyond the objectives of the PPE, as is the onshore wind energy. There are currently 1380 wind farms in France, representing 7950 wind turbines that are spread throughout the country [75].

The objectives set by the PPE to reach 20.1 GW in 2023 and between 35.1 and 44 GW, in terms of the renewable energy to be connected to the grid in 2028, seem challenging to achieve. The dynamics would have to be multiplied by three to reach at least 3 GW connected to the power grid each year to reach these targets. Floating wind power is one of the emerging technologies that can significantly increase the production of renewable electricity in France [76]. According to ADEME (the French Environment and Energy

Management Agency), the potential for electricity production from this technology in metropolitan France is estimated at 155 GW [77]. In France, only one demonstrator is currently in operation and on a relatively modest scale: its capacity attained 2 MW. It concerns Floatgen installed at 22 km off the coast of Le Croisic on the multi-technology sea test site of the École Centrale de Nantes. It has generated electricity since September 2018. Four other projects are in progress, which will reach a total capacity of 115 MW and should be inaugurated in 2023. The main challenge of this technology is to reduce its costs. According to ADEME, the cost should decrease, which could oscillate between 77 and 97 EUR/MWh in 2030 and between 58 and 71 EUR/MWh. This decrease in cost depends on the policies of governments to expand the number of projects worldwide.

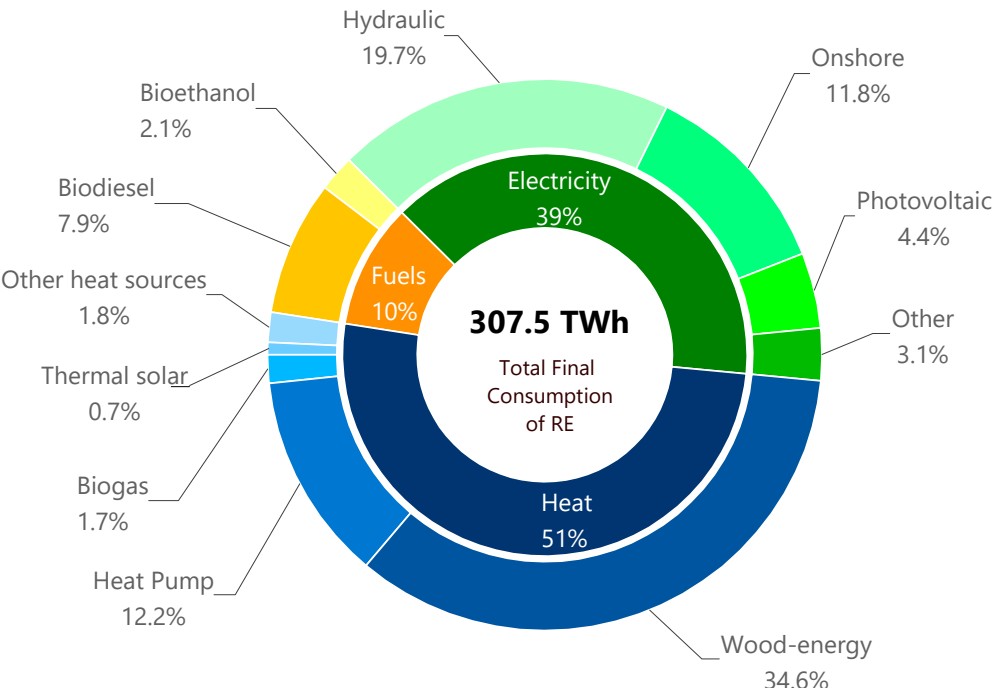

**Figure 1.** The final consumption of renewable energy by sector in TWh (year 2020).

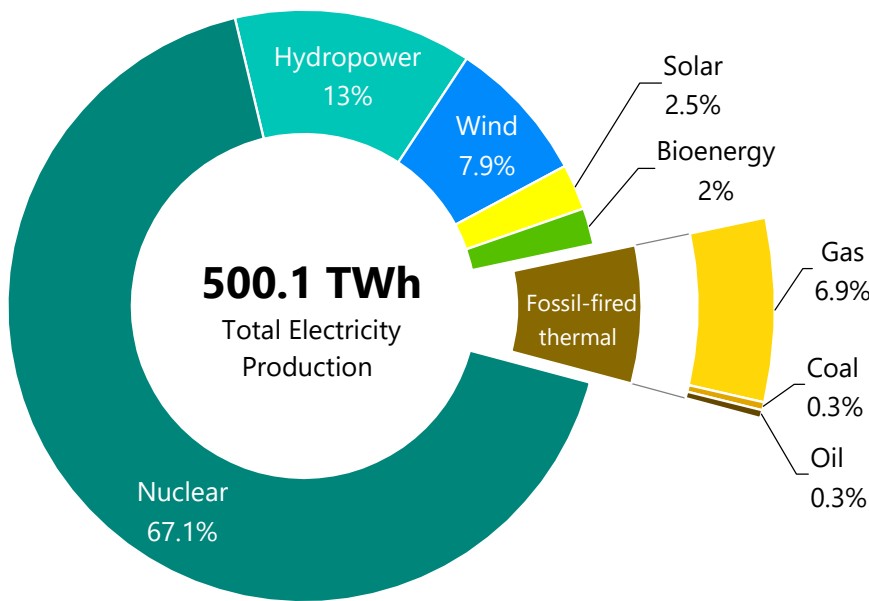

**Figure 2.** The electricity production by source in TWh (year 2020).

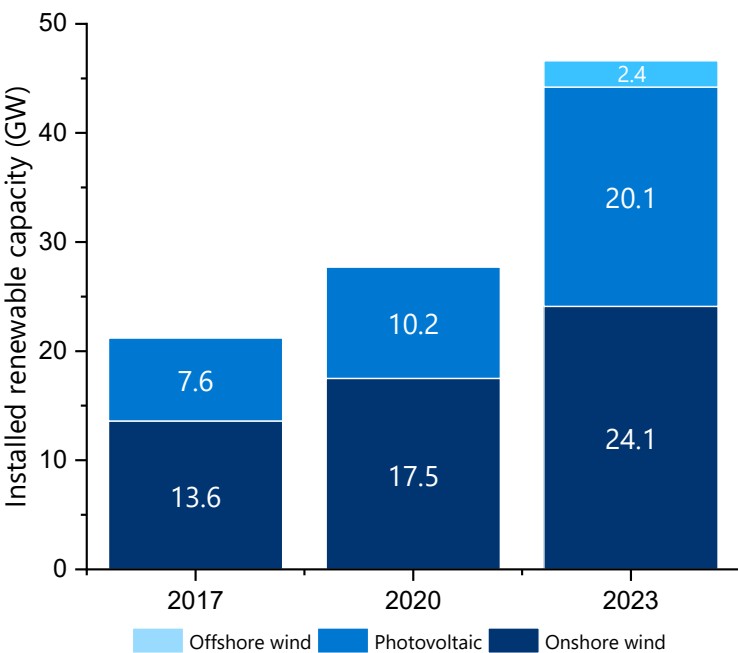

**Figure 3.** The installed renewable capacity (2017–2020) and PPE targets (2023) in GW.

The main renewable sources in France are solid biomass (38.3%), renewable hydropower (20.2%), biofuels (9.9%), wind power (8.8%), and heat pumps (8.7%) [20]. Finally, thermal power plants using gas, oil, or coal are used during peak periods, when other production sources are unable to meet the demand.

France is the leading European producer of bioethanol with a production rate of 32%, twice as much as Germany, which is second. Two thirds of bioethanol are used for fuel, while the rest is used by the beverage, perfume, pharmaceutical, and industrial sectors. France exports 30% of its bioethanol production for fuel, notably to Germany. The French capacity is 19 million hectoliters.

Power plants in France are ordered according to their "merit order". In order to meet the demand for electricity in all circumstances, two further faculties are available:

- France can import electricity from a European country instead of using a national generating facility when the market price is lower than the price of the national facilities. Energy imports are also used in cases where the French generating system cannot meet the demands.
- The hydraulic retention dams can be mobilized to ensure electricity production under the adjustment mechanism.

### 3.2. The Electricity Producers in France

EDF (Electricité de France) continues to be the leading electricity producer in France. EDF's energy mix is strongly dominated by nuclear energy [78]. It is also the leading producer of hydraulic energy in France. It has, indeed, primarily benefited from the obligation of the preferential right enshrined in the law of 16 October 1919 [79]. A total of 322 of the 399 hydraulic plants are conceded to EDF.

In addition to EDF, two other significant players distinguish themselves in a highly concentrated market. The ENGIE group developed the world's second largest power-generation fleet. In France, this group controls the National Company of the Rhône and the Hydro-Electric Company of the Midi (SHEM), which exploit hydroelectric dams [76]. As a result, ENGIE operates a quarter of France's hydroelectric-generation fleet, and more than 50% of its energy mix is composed of renewable energies. E.ON (Energy On) has been a significant actor in the electricity-generation market in France, since the acquisition of Société Nationale d'Electricité et de Thermique (SNET) in 2008. Coal-fired power plants

dominate E.ON's energy mix. E.ON also operates a few gas-fired combined-cycle power plants, as well as a few photovoltaic power plants and wind farms.

### 3.3. The Wholesale Electricity Prices in France

Two wholesale prices can be identified for electricity: electricity purchased under tariffs set by the government, and electricity purchased on wholesale markets [80]. For the latter, the prices vary depending on the product (i.e., peak, base load, or long-term or short-term electricity). In the early 2000s, the wholesale price of electricity was below the regulated tariff price. This difference encouraged many manufacturers, who could choose to leave the regulated rates for market offers, to exercise their eligibility to benefit from more advantageous electricity supply rates. Since 2005–2007, the wholesale market prices for electricity have been higher than the regulated rates. A transitional regulated tariff for market adjustment (TaRTAM) was therefore created in 2006 in order to allow all consumers who exercised their eligibility and wished to return to regulated sales tariffs to have access to a regulated tariff for a maximum period of 2 years [81]. The TaRTAM has not been accessible since 30 June 2010, and was extinguished after the implementation of the Regulated Access Mechanism to Historic Nuclear Electricity (ARENH), provided by the NOME (New Organization of the Electricity Market) law [81]. This mechanism allows consumers to access cheap energy, even if they leave the regulated tariffs, since the alternative supplier can choose to be supplied, either on the wholesale markets or through the ARENH (which is a regulated wholesale tariff) [82]. In practice, the provider arbitrates between these two options. The ARENH, which, since 2011, consists of allowing alternative suppliers to acquire 100 TWh of nuclear electricity at the regulated price of EUR 42 per MWh, is now disputed. The alternative suppliers wish to raise the ARENH limit to 150 TWh, while EDF requests that the ARENH price should be revalued considerably above its current level. At the beginning of 2020, the public authorities submitted a project to the public debate, which plans to replace the ARENH system with a "corridor"-type price mechanism [81]. As the global economy rebounds in 2021, especially the Chinese economy, energy demand has sharply risen. This has led to an explosion in the electricity prices related to the explosion in gas prices. This new situation led the French government to raise the ARENH ceiling for the year 2022, from 100 TWh to 120 TWh.

Practically all nuclear production would be provided on the wholesale market, but the price paid by the supplier, i.e., its customer, must ultimately fluctuate between a ceiling price and a floor price, the two bounds being separated by a maximum of EUR 6 per MWh. The real market is dissociated from a financial market, giving rise to ex-post compensation between EDF and alternative suppliers to obtain this result. The CRE (Energy Regulatory Commission) will set both the ceiling and floor prices on an objective basis. If the spot price of nuclear power is higher than the guaranteed ceiling price, EDF has to pay the difference to the suppliers who bought the nuclear power to supply their customers. If the spot price is lower than the floor price, the alternative suppliers have to pay the difference to EDF.

Electricity markets in France will be fully open to competition, only when the electricity production market is less concentrated and when suppliers have their own production capacities [83]. The ARENH system is therefore only a temporary palliative, due to EDF's current monopoly on electricity production.

EDF is currently heavily indebted and must heavily invest to extend the life of its nuclear power plants and develop renewable energies. The group is currently handicapped by the ARENH mechanism. For these reasons, the Hercule project, renamed "Grand EDF", was proposed in 2018 by the group's management to reorganize EDF. On 28 July 2021, the French government announced that the Hercule project was deferred to the next presidential term.

## 4. The Main Challenges for the Large-Scale Development of Renewable Energies in France

The decarbonization of the energy system is one of the significant challenges of energy transition in France. In the French electricity system, this decarbonization will result in a significant increase in variable renewable energies (such as photovoltaic solar and wind power). These resources are mainly connected to distribution grids that were initially developed to distribute the energy from the transport grids. Electrical systems are therefore faced with the dual challenge of flexibility and decentralization. In the following paragraphs, we present the main tools that are currently being developed in France to meet these challenges.

### 4.1. The Power Grid

Today, electrical energy, which accounts for 45% of primary energy in France, is quasi-entirely transported through electrical grids [76]. Electrical grids are currently performing many functions, ranging from the security of supply to the quality of the energy supplied, and optimizing costs by allowing for consumption and production to expand [84,85]. Power grids are playing a central role in the energy transition today. Their development has an impact on the organization of the electricity industry. As renewable energies become more and more popular, significant investments are required in the networks to move towards a decentralized system [86]. These investments need to be anticipated in order to avoid congestion, as is currently the case in Germany [87].

New uses also require an investment in the power grid to accommodate them. As electric vehicles become increasingly popular, millions of charging points will be connected to distribution networks. This will also require significant investments, since, for example, the rapid recharging of a vehicle in about ten minutes can represent the power demand of a building [88–90].

The decrease in photovoltaic costs and the emergence of virtual energy storage systems [91] can contribute to the multiplication of self-consumers, either individual or collective, and at the neighborhood scale. Networks have to adapt themselves to the societal evolutions related to these new modes of local consumption. The digitalization of power grids (smart grids) is necessary to successfully implement the energy transition [53,61,62]. Indeed, the revolution in connected objects will enable the rapid emergence of smart grids.

The deployment of renewable energies is challenging how the electricity sector has, to date, achieved a supply–demand balance [92]. Today, production no longer adjusts to demand, but is increasingly the opposite, thanks to the development of connected objects and communicating meters. Smart grids can reduce the costs of integrating renewable energies and electric vehicles into the electrical system [85]. In this way, the charging of electric cars can be controlled to match production as closely as possible, and it is likely that, in the future, stationary batteries will be dedicated to facilitating the integration of renewable energies into the network [85]. Energy storage and the possibility of redistributing the energy produced between consumers are two aspects that can also emerge and develop rapidly via smart grids. Blockchain and Artificial Intelligence will also play a crucial role in smart grids. Both concepts can be used to optimize, predict, and archive production, consumption, and energy transactions in real time [93–95].

The French government supports the deployment of smart grids as part of its energy transition policy [74]. France's law on energy transition and green growth explicitly encourages experimentation with smart grids in its regions. To initiate the integration of smart grids at the national level, France conducted experiments with the Linky-communicating meter in 2010–2011. It was evaluated by the CRE, which recommended its generalization in France. The Linky-communicating meter must be installed throughout metropolitan France by 2022.

In 2014, as part of the new industrial France, the government gradually implemented a plan to develop smart grid systems at the national level. Several projects (such as Flexgrid, Smile project, IssyGrid, Nice Smart Valley, and Lyon Smart Community) [96] were

developed to experiment with technological solutions on a large scale, to activate territorial dynamics, and contribute to creating an industrial platform for France's expertise in the field of smart electrical networks.

The Implementation of smart grids raises technological, political, and societal challenges, and therefore presents several issues:

- The implementation of a smart grid requires the installation of smart meters, which can lead to an intrusion into the private life of individuals and, therefore, the generation of personal data subject to the Data Protection Act.
- The implementation of smart grids implies the collaboration of many different actors (e.g., consumers, customers, and public authorities). The importance of consumers' behavior and engagement is crucial to achieving the objectives of the smart grids.

### 4.2. Electricity Interconnections

Thanks to its geographical location at the center of Europe, France has an essential role to play, and is therefore the principal European exporter of electricity. The country exported 84 TWh of electricity to its neighboring countries in 2019 for 28.3 TWh of imports, i.e., a balance of trade equal to 55.7 TWh [97]. Electricity exchanges between two countries can be carried out physically or contractually. Contractual exchanges correspond to the electricity exchanges resulting from commercial agreements between the two countries, whereas physical exchanges correspond to the electricity flows that really transit across the interconnection lines [98]. Under the PPE, France's overall interconnection capacity in 2019 will be 17.4 GW for export and 12.5 GW for import.

The benefits of electrical interconnections include the ability to compensate for the variability of intermittent renewable energies (wind and photovoltaic) by combining them, and to reduce the costs associated with their integration by mutualizing the reserves and sources of flexibility [99,100]. Every year, RTE draws up a ten-year network development plan (SDDR), which is submitted for approval to the CRE. The RTE's 2019 SDDR highlighted the objective of developing France's interconnection capacity, which could double in 15 years from 15 GW today to 30 GW by 2035. The CRE examined this objective on 30 July 2020. In its deliberation, the CRE noted that France's objective of reducing the share of nuclear power by 2035 implies the integration of more and more renewable energies, requiring a structural adaptation of the transmission network. This will benefit the development of electrical interconnections. Today, with an average import/export interconnection capacity of 15 GW, France has an 11.5% interconnection with its neighbors. In 2030, it should exceed 26 GW of interconnection to reach at least 16.5% [20].

### 4.3. Hydrogen

France's Multi-Year Energy Program aims to increase renewable electricity production from 102 to 113 gigawatts in 2028, twice as much as in 2017. Power to hydrogen (P2H$_2$) is among France's tools to support this transition to renewable energies. It is an alternative that could provide flexibility to an electrical system mainly based on renewable energies [101,102]. Power to hydrogen has the potential to prevent the waste of available low-carbon energy, either by consuming the "surplus" of renewable production during off-peak periods, or benefiting from the "surplus" of available nuclear production, which is due to the modulation of nuclear power plants to remain synchronized with the changes in residual demands [103–105].

The key energy conversion process needed to generate hydrogen is water electrolysis. There are three types of water electrolysis technologies: proton exchange membrane (PEM) electrolysis [106,107], solid oxide electrolysis (SOE) [108,109], and alkaline water electrolysis (AWE) [110]. Today, hydrogen production using electrolysis is only 6% in France. GRT gaz, Teréga, EDF, Air Liquid, Engie, H2Gen, McPhy, AFC Energy, Hynamics, and Lhyfe are among the main actors of electrolysis hydrogen production in France. Many research water electrolysis projects are in progress, with the following objectives:

- Reduce the investment cost of technologies;
- Enlarge the power modulation range of the electrolyzers;
- Increase the operating temperature of the electrolyzers.

The hydrogen produced can be:

- Injected (or synthetic methane derived from hydrogen and carbon dioxide) into the gas network, usually described as "power to gas". The adaptation of gas distribution networks to hydrogen transport needs to be explored and considered [111,112].
- Stored for use in transport or industry.

Storage and distribution methods are more or less advantageous and costly, depending on the volume, pressure, and storage duration [113–115]. For consumers below 50 $m^3$/h, it is conditioned in a gaseous form in bottles and transported by semi-trailers (from 3000 to 6000 $m^3$ of $H_2$ at 200 bars). Hydrogen gas is stored at 700 bars in fuel cell vehicles for mobility. Since the energy density of hydrogen by volume is low, it is necessary to compress it at a very high pressure to reduce its storage size. The hydrogen compression represents about 7% of its calorific value to transition from 1 to 200 bars and 10% to reach 700 bars. Air Liquide, TotalEnergies, $H_2$Logic, Faurecia, Michelin, Toyota, Symbio, and Plug Power are among the leading actors of hydrogen mobility in France.

Hydrogen can also be stored in a liquid or solid form. Liquid hydrogen storage is not economical for extended storage times due to the gradual decrease in the volume stored due to the boil-off effect [116]. In fact, the storage tank absorbs heat and thus vaporizes the liquid hydrogen, which must be vented to avoid creating overpressure in the tanks. Solid storage in metal hydrides has the advantage of a higher storage density [117]. Hydrogen can be stored and removed from storage by thermal input at an atmospheric pressure, thus avoiding conditioning costs (such as compression and liquefaction). On the other hand, hydrides are penalized by their weight and the thermal management required for the adsorption and desorption of hydrogen. Several research projects are currently underway to improve hydrogen storage technologies.

Gas companies use hydrogen pipeline networks, when consumption exceeds several tens of thousands of $m^3$/h of $H_2$. The Western European network is the largest globally, with more than 1600 km of pipelines and 50,000 $m^3$/h of hydrogen at 100 bars [118,119].

Power to hydrogen can provide three levels of flexibility (via fuel cells) to the power grid: flexibility in terms of time, place, and end use [105,120]. Power to hydrogen's time flexibility is that it is capable of adapting the scheduling of the use of power and the generation of hydrogen. If a hydrogen production plant is combined with a hydrogen storage system, the timing of the production process can be fully accommodated to the fluctuations in electricity prices, while the storage can provide inexpensive hydrogen to consumers.

With the power to hydrogen technologies, France can significantly reduce its greenhouse gas emissions in the industrial, automotive, and tertiary sectors, which, today, represent the most energy-intensive sectors [121]. Several experimentations on power to hydrogen technologies (Myrte, Jupiter, and Ghryd) have been launched in recent years in France (see Table 2) [122]. Among these demonstrators, the MYRTE platform is particularly interesting, both for its primary solar power level and because it was the first to be connected to the power grid and has been the subject of business model experimentation [122]. Although the efficiencies of the hydrogen chain are relatively low (35%), such a solution is of interest in the field of energy storage, where STEPS and CAES cannot be installed, or in hybridized solutions with other forms of storage.

**Table 2.** The main power-to-hydrogen projects in France.

| Project | Region | Industrial Partners | Main Targets | Budget | Status |
|---|---|---|---|---|---|
| **GRHYD** | Hauts-de-France | Engie, Areva, CEA, Engie Ineo, GRDF, McPhy, etc. | • PtG, recycling mixure for NGV buses and 100 housing unit.<br>• Distribution injection (up to 20% H2) | €15 M | In operation |
| **Zero Emission Valley** | Auvergne-Rhône-Alpes | AuRA, SME, Ataway, Michelin, Engie | • 15 electrolysers, 20 charging station and 1000 vehicles | €70 M | Under construction |
| **HyGreen provence** | Provence-Alpes-Côte d'Azur | DLVA agglomeration community, Geometane, PACA | • PtG for mobility (bus + train) and injection, storage in salt caverns<br>• Planned transmission injection | €300 M | In the study |
| **Jupiter 1000** | Fos sur Mer | GRTgaz, CEA, Atmostat, RTE, Teréga, McPhy, Leroux&Lotz. | • PtG, Catalytic methanation with industrial $CO_2$ (25 Nm)$^2$/h | €30 M | Under construction |
| **Hycaunais** | Bourgogne-Franche-Comté | Storengy, Areva H2Gen, Engie, Electrochaea, Waga Energy, etc. | • PtG, methanisation coupling and biological methanation,<br>• Planned gas network injection | €9 M | In the study |
| **MéthyCentre** | Centre-Val de Loire | Storengy, Atmostat, Areva H2Gen, CEA, Prodeval, ADEME | • PtG, catalytic methanation with $CO_2$ from biogas,<br>• Injection of gas networks | €12 M | In the study |

In 2020, the French government announced a budget of EUR 7 billion to implement a national strategy for developing decarbonized hydrogen. Three axes will be developed within the context of this program:

- Decarbonize the industry by creating a French electrolysis sector.
- Develop heavy mobility with decarbonized hydrogen.
- Support research, innovation, and skills development to promote future uses.

Another element that can contribute to accelerating the emergence of power-to-hydrogen technology in France is the implementation by French industrial company "Hydrogène de France (HDF)" of a high-power fuel-cell production plant (more than 1 MW) [123]. This factory will be the first in the world to produce high-power fuel cells based on PEM (Proton Exchange Membrane) technology. The industrialization and standardization of these systems can reduce their cost and significantly improve their reliability and maintenance. HDF's vision is based on the development of hydrogen as a fuel and electricity storage vector.

Hydrogen is on the way to becoming an energy carrier, in the same way as electricity or heat (see Figure 4). After an initial development phase dedicated to transportation-oriented technologies, this vector has gradually taken on an increasingly central role in a context where the ability to store and transport energy is becoming a priority. The electric vector can be used for any purpose, whether stationary or mobile. The technologies for generating, storing, and converting hydrogen are now (mostly) under operation.

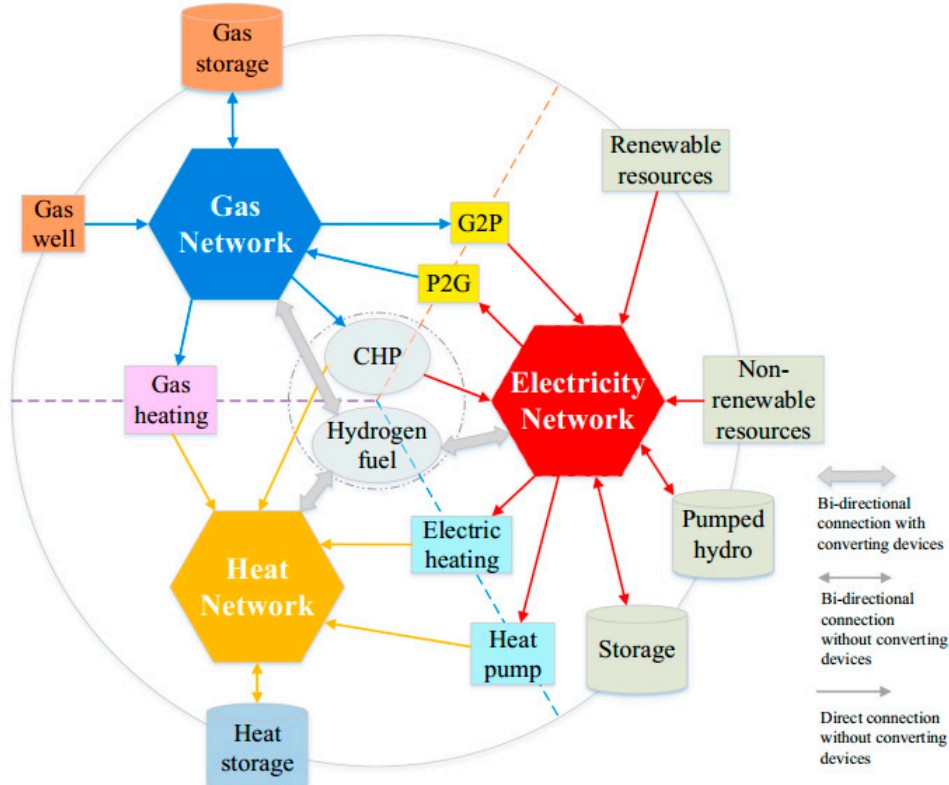

**Figure 4.** The links between energy systems and associated technologies (Reprinted with permission from ref. [124]).

### 4.4. Electric Vehicles

Battery-powered electric vehicles appear to be the future solution to ensure mobility independent of fossil energy resources and non-emitting $CO_2$ during use [125,126]. In 2021, Plug-in EV sales surpassed 6.75 million units in 2021, 108% more than in 2020. EVs accounted for 8.3% of worldwide light vehicle sales in 2021, up from 4.2% in 2020 (see Figure 5) [127,128]. Compared to other regions, Europe's EV sector saw significantly more growth, and is expected to hold 27% of the global EV market by 2030 [129].

To increase the penetration of battery electric cars, three barriers need to be addressed [126,130,131]:

- The autonomy that will be achieved by increasing the performance of the batteries and the use of fast charging [132–134].
- Battery prices. The price of lithium-ion batteries decreased by 85% between 2010 and 2018, as a result of the technical advances in the cathode, cell, and package, which improved energy density, and further decreases and performance improvements are expected as a result of increasing industrialization, so that some analysts predict a price of USD 100/kWh within a decade or so [135,136]. Volume effects (a significant increase in the number of vehicles and, therefore, the number of batteries manufactured) will allow for price reductions. For example, in 2020, the two French groups PSA and Total, through their subsidiary Saft, announced the creation of the Automotive Cells Company (ACC) to produce lithium-ion batteries for electric vehicles.
- The lack of availability of recharging points also limits the deployment of electric vehicles. According to [137], most light-duty electric vehicle chargers in 2019 were private chargers in homes, workplaces, and multi-dwelling buildings, while publicly, accessible chargers accounted only for 12% of total chargers, most of which were slow chargers (See Figure 6).

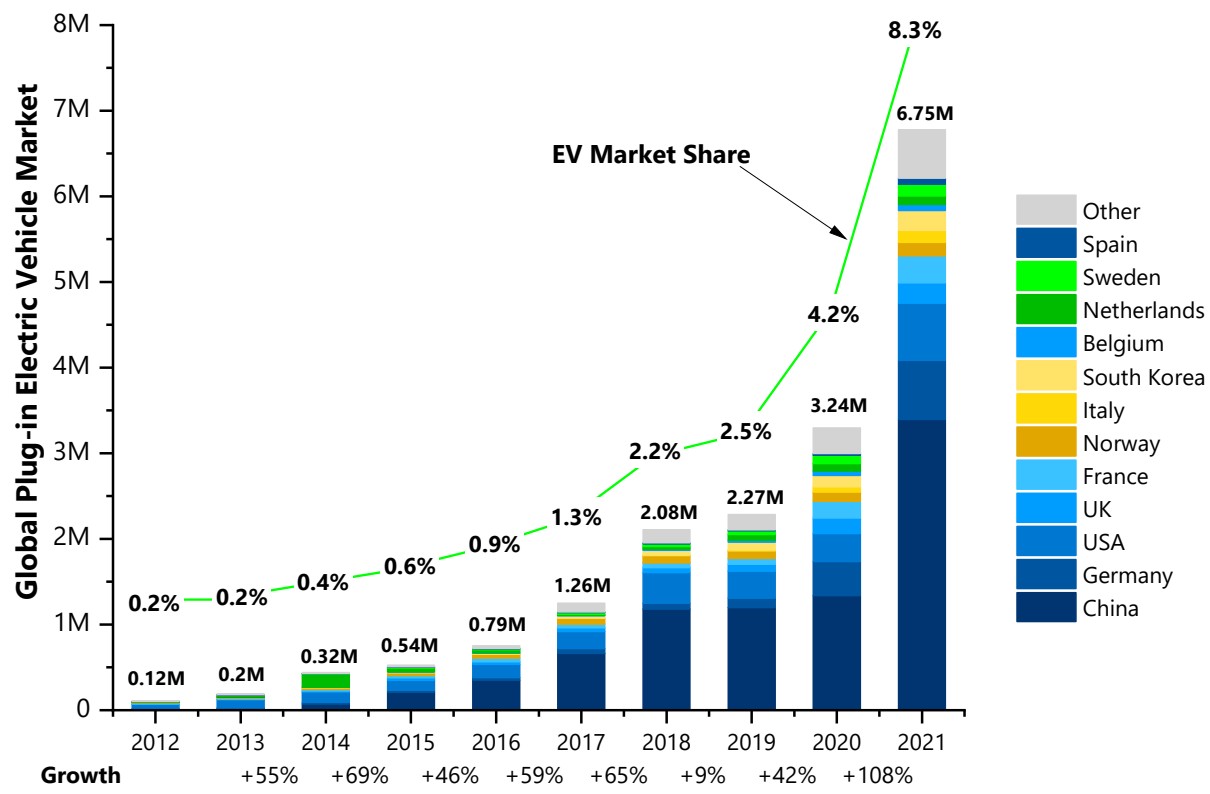

**Figure 5.** The global plug-in EV sales (Adapted from ref. [128]).

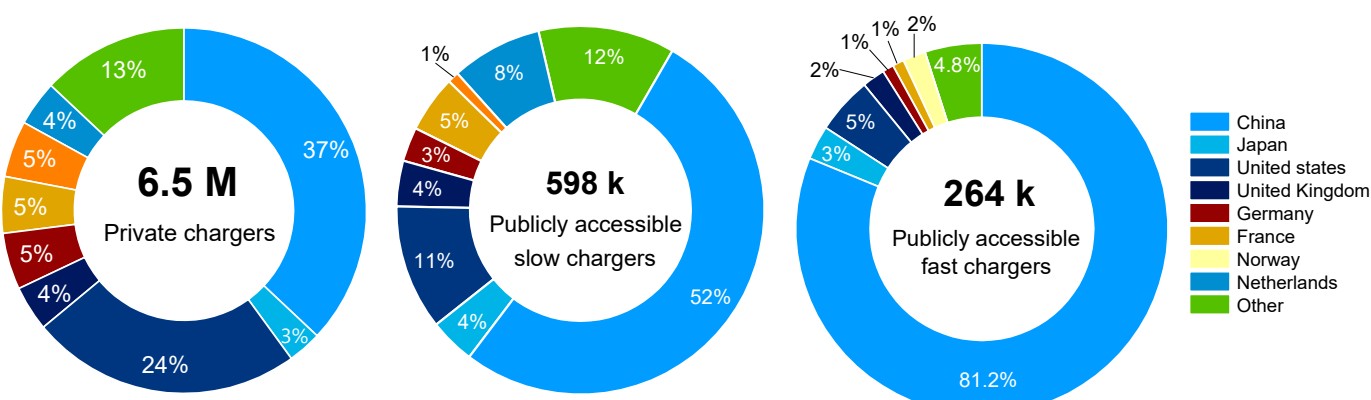

**Figure 6.** The privately and publicly accessible chargers by country, 2019 (Adapted with permission from ref. [137]).

Electric vehicles with batteries need to be connected to the grid for charging [138]. Recharging a large number of electric vehicles from the same electrical network will significantly impact the network due to the consumption growth induced, and on the quality of the electricity supplied by the network [139]. The power demand of vehicle loads can high huge peaks in the network. The peak power needed may become a significant problem for the network and in terms of emissions, as this peak power is highly carbonized. Therefore, it is necessary to avoid this load at peak periods, shift it during periods of low consumption, or, during the overproduction of variable renewable energy, the load must be controlled [140]. The following is a major challenge of intelligent charge: to move the load as far as possible in the consumption troughs by allowing for the automatic modulation of the load power [88,141].

The forecast in France is for 2 million electric vehicles to be in use by 2025, and 5 million by 2030. By 2030, this will require 7 million charging stations, most of which will be connected to users' homes. Three types of recharging power are currently in use [142]:

- Normal recharging (a 3 kVA, 16 A, terminal connected to a single-phase network) allowing for a recharging time of about 8 h.
- Accelerated recharging (a 22 kVA, 32 A, terminal connected to a three-phase network) allowing for a recharging time of about 2 h.
- The fast recharge (a 43 kVA, 63 A, connected to a three-phase network) allowing for a recharge in 20 min. This type of recharging exists in direct or alternating currents.

According to the type of recharging power implemented, the impact on the electrical network will be different. For example, recharging from a terminal connected to a single-phase network can impact the current and voltage imbalances of the network. Accelerated or fast recharging will have a more significant impact on a possible overload of the network. In addition, increasing the penetration of EVs brings new challenges to the distribution system operator (DSO) and requires new approaches and strategies to integrate these active distributed resources in sizeable quantities as well as the aggregator [143,144] (see Figure 7). According to [143,145–147], the aggregator is defined as "*an entity that creates a group of E.V.s capable of exploiting their bidirectional power capability to act as a distributed energy resource that can impact the grid positively*". It is an upstream link to various network entities.

Through this concept, using an EV battery as a storage medium for the network when it is connected can be a possible solution for load balancing and for reducing the ownership cost of an EV [144]. Additionally, with the development of intelligent and bidirectional charging stations, electric vehicles can transfer energy to the power grid (V2G), to EVs (V2V), and to the residential (V2H) or tertiary (V2B) buildings to which they are connected [148–150]. These concepts allow buildings to be deleted from the power grid during critical peak periods, and thus contributes to its reliability.

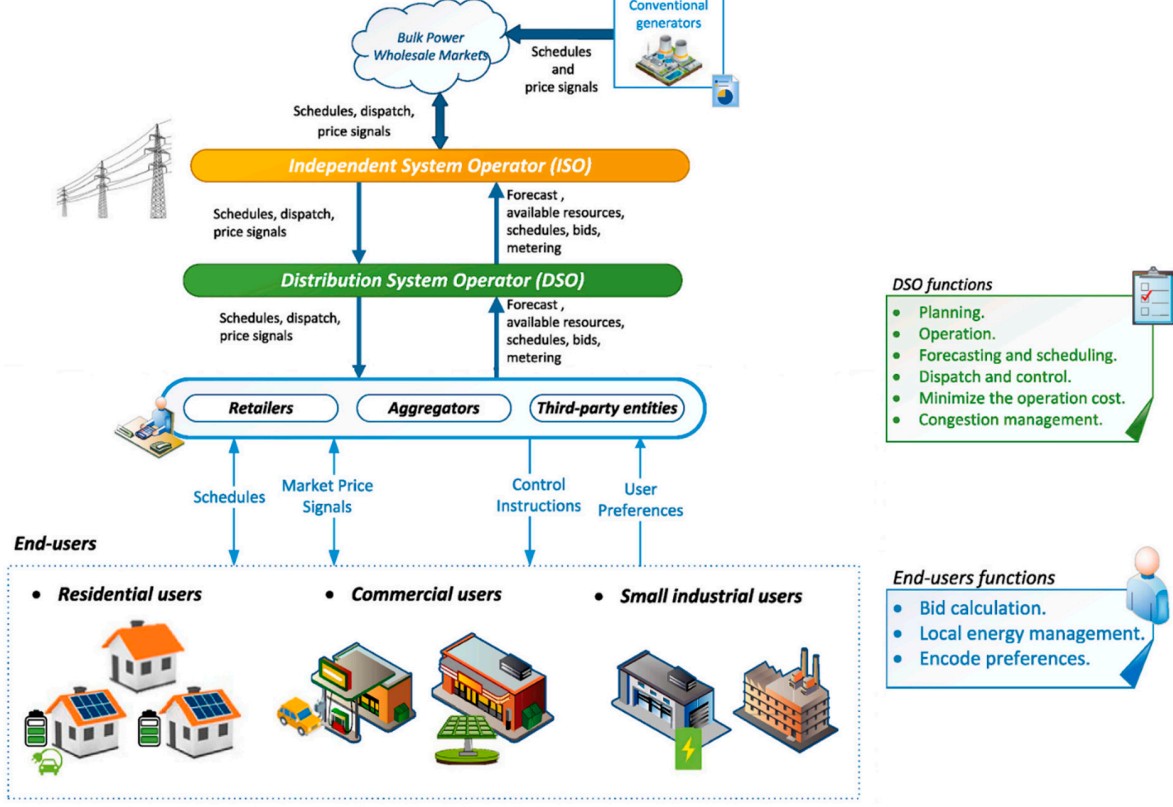

**Figure 7.** A schematic diagram of the distribution system operator (DSO) (Reprinted with permission from ref. [151]).

In France, DREEV, which is a set up between EDF and NUVVE, is one of the companies working in this area in France. It is already deployed the V2G terminals in the Bordeaux region, permitting the company to save up to EUR 20 per vehicle per month [152]. DREEV also operates in several fields, including smart charge management, and the implementation of energy flexibility systems to balance supply and demand. Different providers will provide these solutions, particularly IZIVIA, a wholly owned EDF Group subsidiary [152].

Otherwise, the decrease in the cost of batteries can lead to a significant transformation of the electrical system, in which the adjustment between supply and demand had to be made in real time by modulating the production. If we combine the existing major means of electrical storage with hydraulics, the rise of the electric vehicle leads to the prospect of a large availability of Li-ion batteries of more or less small capacities, which can be used to optimize self-consumption and, eventually, to equilibrate the national grid.

The fast, wide spread of electric vehicles is consequently coupled with an increase in Lithium-ion-battery demand. Over the next five years, major battery manufacturers will invest over USD 50 billion to increase Li-ion battery production capacities, which are projected to reach 1.2 TWh by 2020 [153]. Accordingly, growing attention is being given to the availability of lithium (Li) [154–156]. To investigate the role of Li in the future energy system, a study is provided in [136], in which the long-term availability of Li, in the case of the significant increase in demand for rechargeable lithium-ion batteries for power supply and transport, with a very high share of renewable energy is explored. In this study, Li is essential for a sustainable energy transition. Indeed, ensuring sustainable supply and demand for Li over the coming 20 years depends on well-established recycling systems, V2G integration, and lower Li intensity in transport services [136,157,158].

### 4.5. District Heating

Many historians consider France to be among the first countries to set up a district heating system (at Chaudes-Aigues, a small town near Lyon in the 14th century [159]), and, since then, it has become a "country of learning about D.H." that is compared to Northern and Eastern European countries. According to the Euroheat and Power (EHP) data for 26 European countries, there are approximately 200,000 km of district heating [160]. France ranks twentieth in the classification of countries using heating networks for heating needs. According to the 2018 data from the Observatory of Heating and Cooling Networks, three million people are connected to a heating network in the residential sector and three million in the tertiary sector [161]. Heat accounts for almost half of the final energy consumption, i.e., 751 TWh. In France, district heating operators have been converting the primary energy used to supply heating and cooling plants for more than ten years. To date, with the 4th generation of district heating (see Figure 8), renewable and recovery energies represent the second-largest source of heat (155 TWh, or 21%), after gas [159]. The first renewable heat source is solid biomass (79%). In 10 years, renewable and recovery energies have thus increased from 27% to 57.1% of the energy used by district heating.

In France, the heating fund was doubled between 2015 and 2017 to reach EUR 420 million to encourage district heating to use renewable energies (e.g., biomass, geothermal, and solar thermal) [162]. In January 2020, as part of the PPE, France fixed the objective of achieving a quantity of renewable heat and cold and recovery delivered by the district heating between 32.4 TWh and 38.7 TWh in 2028, i.e., an increase of 50 to 100% compared to the current rate of development. Encouraging local, renewable, or recovered energy sources in the energy mix of heating networks is critical in limiting greenhouse gas emissions. In France, according to the ADEME, waste heat from the industry represents a source of 52.9 TWh rejected in the form of waste heat.

Fifth-generation district heating is currently emerging as a solution to today's environmental challenges [163,164]. Indeed, they tend towards 100% locally supplied renewable and recuperated energy. They are based on a tempered hot-water loop [165], which allows taking advantage of renewable energies and low-temperature recovery, such as geothermal energy, or even waste heat from industrial or data centers. This closed-loop, with a temper-

ature approaching that of the ground, coupled with heat pumps, allows for the recovery of heat produced by cold consumers (and vice versa), thus avoiding catastrophic thermal energy production losses. Finally, the advanced demand management is implemented to avoid consumption peaks and thus use more carbon-intensive energies via deletion or thermal storage systems.

Beyond the desire to use renewable and recuperative energies as much as possible, the technical reality often leads to plan backup and emergency productions that regularly use conventional energies [166,167]. It is necessary to define the entire district heating during the conception phase: primary, auxiliary and backup production, distribution, and regulation, without forgetting the impact on the secondary networks of the buildings supplied. District heating is considered relevant if it is designed to mutualize energy needs, prioritize the use of renewable energies, guarantee a competitive investment and consumer cost, and be sustainable [168]. Today, district heating is moving towards artificial intelligence solutions by equipping itself with intelligent technologies, communicating sensors, and thermal energy storage units, and by interacting with other energy networks (e.g., electric and gas) [169,170]. Thanks to these new technologies, smart district heating operators can finely predict network demands, detect anomalies, and calculate trade-offs on the energy mix.

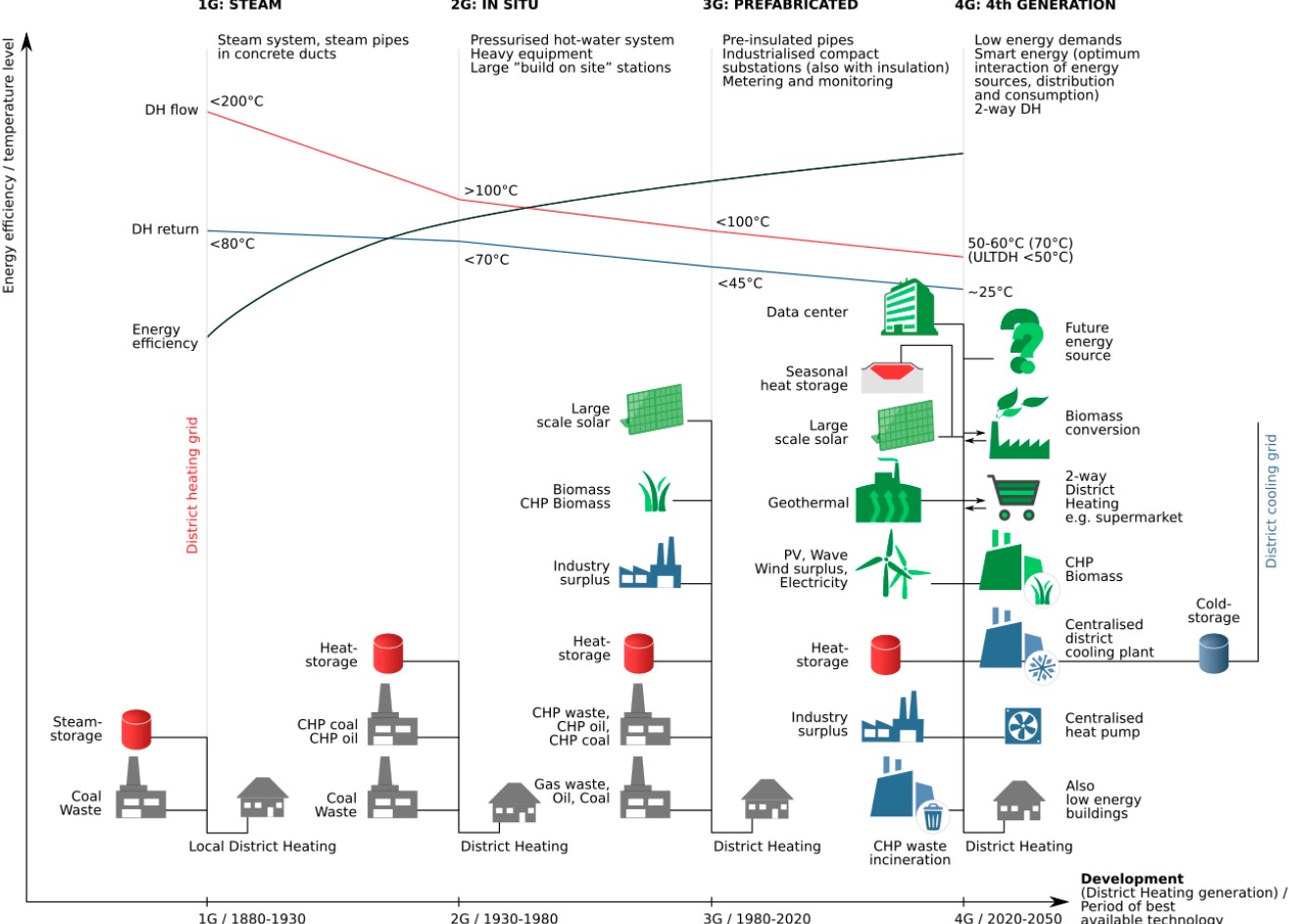

**Figure 8.** An illustration of the concept of 4th-generation district heating, in comparison to the previous three generations (Reprinted with permission from ref. [171]).

*4.6. Heat Fund*

In order to achieve France's 2050 carbon neutrality goals and increase the share of renewable energy to 33% of the gross final energy consumption by 2030, the decarbonization of heat production plays a significant role. Heat represents nearly half of France's energy consumption, and less than a quarter of it is currently produced from renewable energy sources [48,172]. Established by the French government in 2009, the Heat Fund, managed by ADEME, aims to encourage the replacement of installations that consume fossil fuels by the installation of equipment for the production of renewable heat and cold using biomass, geothermal energy, solar energy, biogas, and recovered energy, coupled with heat and cold networks. It is intended for collective housing, communities, and companies [173]. Over the past 12 years, the Heat Fund has continued to evolve and has helped more than 6000 renewable energy and recovery (RE&R) installations with EUR 2.6 billion of aid, generating EUR 9.4 billion of investment. This represents nearly 36 TWh/year of the additional production of RE&R. Despite the unprecedented health situation (COVID-19), the year 2020 balance sheet is sharply up. Projects that have received investment aid from the Heat Fund will generate 4 TWh/year of additional renewable heat. Wood energy contributes the most to this renewable heat production, followed by methanization (28%) and waste heat (5%), and then geothermal energy (9%) [174]. These aids are remarkably efficient, with a public aid amount of EUR 4.4 per MWh produced over 20 years. However, to reach the objectives set by the PPE, the annual production of additional renewable heat should be doubled, compared to the trend observed in recent years.

*4.7. Energy Storage*

In France, the seasonal storage of electricity is assured by large hydroelectric dams, while weekly sluices and run-of-river structures regulate hydropower by making it a flexible renewable energy source [66,175]. On the demand side, the storage of domestic hot water (power to heat) in the residential and tertiary sectors represents a vital management tool on the daily horizon, making it possible to smooth the load curves and reduce the use of peak resources at a very low cost [176]. Several mechanisms were implemented in order to associate this type of thermal storage with electricity-based (heat or cold) production systems (i.e., heat pumps) to provide essential services to the French electrical system. The development of district heating can also designate an increasingly important role to this type of storage.

Pumped energy transfer stations provide centralized electricity storage in France [177]. It represents around 1% of the energy produced in France and targets high-value-added services. France has 6 large STEPs, commissioned between 1976 and 1987. They represent a power of 4.2 GW in pumping and 4.9 GW in turbining [65].

The emergence of variable renewable electricity generation, such as wind and photovoltaic, which, unlike hydropower, requires the development of new energy-storage technologies. At present, the French government focuses on three types of storage:

- On-board storage via the development of electric cars [88,142]. The prospects for this type of storage seem favorable, since 2.6 million electric cars were registered worldwide in 2019, with a 48% annual increase since 2015. This growth should continue in the coming years, including in France, encouraged by the decrease in battery prices. In 2020, Total, through its subsidiary Saft, and PSA with Opel announced their intention to combine their expertise to develop an electric vehicle battery production business in Europe and their intention to create a joint company called ACC (Automotive Cell Company) for this purpose [178]. This manufacturing plant can accelerate the electric car deployment in France and Europe.
- Stationary storage with batteries for power adjustment [157,179,180]: the worldwide installed capacity of stationary batteries did not exceed 1 G.W. before 2015; it reached 10 G.W. in 2019 and is expected to proliferate over the next few years. The drop in the price of batteries for electric vehicles is not necessarily ipso facto transferable to stationary storage. In the first case, the main R&D effort is focused on increasing

autonomy, while, in the second case, it is aimed more at increasing the number of cycles in nominal operations.

- Self-consumption and virtual storage mechanisms [181–183]: unfortunately, the volumes of storage remain minimal in France, because the low price of kWh compared to other European countries does not facilitate the development of this kind of storage.

Today's real challenge is to propose the most appropriate storage technology solution for the local context. Nevertheless, to find its place among the solutions deployed, the storage system will have to demonstrate its technical relevance and prove its economic profitability [184].

*4.8. Energy Efficiency*

Energy efficiency policies in France evolve from year to year. Even if their modalities may vary with the succession of governments, one can observe continuity in assessing the challenges, the objectives, and the types of instruments deployed.

The Energy Orientation Law of 13 July 2005, known as the POPE law [185], set a target of 2% annual reduction in final energy intensity for 2015, then 2.5% in 2030, while it was about 0.8% on average over the past 20 years. This law placed energy demand management as the first axis of the energy policy.

France's energy-efficiency policy is based on several measures, some of which are transverse in nature, while others are specific to each sector.

The most critical transversal measures promoted by the French government are:

- The energy-saving certificates scheme [186], which was created by the law of 13 July 2005. According to this mechanism, the public authorities impose to energy suppliers the obligation to achieve energy savings for their customers through the active promotion of energy efficiency.
- Support for the development of energy performance contracts.
- Support for the most energy-efficient products, through regulatory and financial measures, such as the sustainable development tax credit.
- The future investment program created by the French government in 2010 to accelerate the market launch of innovative solutions, particularly in the area of energy transition [187].

The building sector is the second-largest emitter of greenhouse gases in France, being responsible for up to 25% of $CO_2$ emissions in 2019 [188]. The thermal regulation and the energy-performance diagnosis are the two key measures of the French policy in this sector. New regulations concerning the energy performance of new buildings have been adopted approximately every ten years, since the 1970s. The year 2020 marked a real change, with the development of RE2020 (Environmental Regulation 2020), which succeeds the R.T. 2012 (Thermal Regulation 2012) [189], and in which we no longer refer only to energy performance, but also to environmental performance [190].

Recently, the French government launched a major plan for the energy renovation of residential and tertiary buildings [188]. The aim is to achieve carbon neutrality by 2050, while pursuing a social objective of reducing energy precarity. The plan includes 4 major axes:

- Make the energy renovation of buildings a national priority.
- Massify the renovation of housing and fight against energy insecurity.
- Accelerate the renovation and energy savings of tertiary buildings.
- Reinforce skills and innovation.

In the context of constrained public finances, the government is generally faced with two choices: to support a large number of renovations that are not very ambitious in terms of energy efficiency, or to support a few very ambitious renovations. The ambition of the EnergieSprong [191] program (deployed in France since 2016, financed by national and European funding, and supervised since 2019 by the public authorities) is to embark on a third path: to support a large number of very ambitious, but less expensive, energy-

efficiency renovations to combat energy precarity. However, several challenges are posed to launch the renovation plan and drastically lower their cost, despite this program.

The French government introduced another "Eco Energie Tertiaire" mechanism on 23 July 2019, to accelerate the renovation of the tertiary sector [192]. This mechanism targets buildings larger than 1000 m$^2$; these must gradually reduce their energy consumption by 40% in 2030, 50% in 2040, and 60% in 2050. Two criteria expressed in kWh/m$^2$/year are defined, and at least one of them must be achieved:

- An "absolute value" criterion defined according to the type of activity;
- a "relative value" criterion defined in relation to a reference year chosen by the taxpayer.

The monitoring of this mechanism is ensured by the OPERAT platform [193] that all those subject to it must fill in by 30 September 2022 (this is a one-year postponement compared to the first date), with information on the building, the history of energy consumption (2010 to 2020), and define a reference year. Then, before September 30 of each year, they must provide the previous year's consumption.

## 5. Conclusions and Policy Implications

Many scenarios were developed in recent years to simulate the evolution of the French energy mix between the present day and 2050. Among these scenarios, two groups can be distinguished: the first corresponds to the scenarios called "stability"; the projections up to 2050 are based on a continuation of the current developments and practices. The second family, which is the most realistic, includes "rupture" scenarios, which consider that the radical changes in our consumption practices must rapidly occur. These changes require considerable efforts on the part of the public authorities and French citizens to achieve an energy situation that is very different from the one that exists today. Based on this assessment, and therefore on the energy transition policy planned by the French government, it is necessary to draw a certain number of conclusions regarding the challenges to be met in the coming years:

- Extending the lifespan of nuclear power plants requires an irreproachable reinforcement of safety mechanisms. It is crucial to pursue the development of educational programs regarding the safety and dismantlement of nuclear power plants to strengthen international expertise in this sector.
- The decline in the share of nuclear power must be accompanied by a significant promotion of renewable energy production by acquiring appropriate levels of flexibility. Otherwise, electricity production from gas will be used as a substitute.
- Energy storage, which can provide several flexible services at different points of the power grid and can link with other energy networks (e.g., gas, heat, and mobility) and thus use their flexibility, and should be developed according to local needs. The other challenges involve optimizing the economic model and minimizing the environmental impact of energy storage technologies by limiting the amount of materials required, reducing the volumes and surface areas used, and promoting recycling.
- For the transport sector, whatever the future of batteries or hydrogen fuel cells may be, and regardless of the technological progress, the development of these systems will be largely linked to the electrical energy or hydrogen distribution network. Safety and environmental aspects must be taken into account when choosing such a form of technology.
- The emergence and expansion of battery electric cars in France requires the development of fast-charging stations. Rapid charging is not possible without significant electrical power. Almost all of the existing charging stations in France provide less than 50 kW, but it will be necessary to increase them to 100 or even 150 kW. The high-power network is particularly insufficient on major roads where electric vehicles need to be recharged frequently and, if possible, quickly. Standards are another major problem, which are often incompatible, such as the payment methods, as well as the subscriptions for certain networks and in certain regions. These standards must be harmonized nationally to allow users to benefit from them.

- Energy efficiency is marked by the diversity of the systems involved and by the inertia required to substitute the infrastructures, installations, and equipment already in place, as well as the markets that contribute to them. Performing annual energy renovations in housing not only requests investment capacities shared between users and various financial incentives, but also assumes that competent installers are available in number to perform these renovations at the desired pace. The analysis of the renovation market reveals that the combination of these determining factors is not an easy task.
- The energy transition in France requires vast quantities of metals (sometimes precious) and minerals. A national strategy is needed to reduce the dependence on third-world countries by improving the efficient use and recycling of these materials and promoting a responsible form of supply.
- A national strategy for waste management is essential to support the energy transition in France. The actions that policy makers can implement through this strategy include the development of and the increase in France's capacity to produce recycled waste that directly meets the energy needs.
- The energy transition goals can be accomplished only through the extensive deployment of services using smart technologies that significantly reduce our carbon footprint. In order to adhere to such services, users have to be assured that they have control and ownership of their personal data.
- The energy transition will not occur without a behavioral change and an evolution of our lifestyles. Nowadays, we know that technological developments and technical approaches alone are not enough to ensure a successful energy transition. It is essential to consider human and social dynamics in order to change behavior, and especially lifestyles, at the local level.

**Author Contributions:** Conceptualization, B.E.L., E.S. and B.L.; Formal analysis, B.E.L. and E.S.; Investigation, B.E.L., E.S., B.L. and T.K.; Methodology, B.E.L., E.S., B.L. and T.K.; Project administration, T.K.; Resources, B.E.L. and E.S.; Supervision, T.K.; Validation, Y.C. and T.K.; Visualization, Y.C.; Writing—original draft, B.E.L., B.L., Y.C. and T.K.; Writing—review & editing, B.L., Y.C. and T.K. All authors have read and agreed to the published version of the manuscript.

**Funding:** This research received no external funding.

**Institutional Review Board Statement:** Not applicable.

**Informed Consent Statement:** Not applicable.

**Data Availability Statement:** Not applicable.

**Acknowledgments:** This research was carried under the framework of E2S UPPA supported by the "Investissements d'Avenir" French programme managed by ANR (ANR-16-IDEX-0002).

**Conflicts of Interest:** The authors declare that they have no competing interests.

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
