# Peer review of "Energy Transition in France"

_sustainability, doi:10.3390/su14105818_

Round 1

Reviewer 1 Report

Line 40 - I think if you are solely talking about crude oil prices, then that should be specified rather than the ambiguous "hydrocarbon". 

Line 42 - shale gas and shale oil? I'm not familiar with "schist oil" being used in the U.S. parlance

Line 68-70 - If these are IEA STEPS and SDS scenarios, then these need to be mentioned more clearly. Right now I don't really know where these scenarios are coming from and I can only surmise the IEA connection based on Line 71.

Line 70 - typo. should be double quotation instead of >>

Line 96-99 - I am not sure if this is the objective the title of the paper should be "Energy Transition Policies in France".

While I appreciate the level of detail and historical notes, Sections 1 and 2 are too long for a topic that is expected to focus on the energy transition in France. Some context is useful for sure, e.g. the latest EC regulations and the the importance of overarching  global trends such as the 2C target and fossil fuel prices to the French energy transition. However, spending 4 pages to get to the meat of the discussion seems a bit much.

Line 194 - I don't know what triptych means, it's not a standard English word. I don't know what sobriety means in the context of energy mix either. There should be a more standard word for this.

Line 198 - What is latter? The electricity system? This can be phrased better with something like "According to this scenario, 100% renewable generation can be achieved without destabilizing the system. This can be achieved through..."

Line 200 - Again, no clue what sobriety is supposed to mean.

Line 202 - Generally, when the issue of preserving raw materials is being mentioned, it needs to be made clear are we talking about French resources only or global resources. Because most of global lithium comes from Australia, Chile or China.

Line 207 - so does sobriety mean reduced energy demand, then that should be specified explicitly. 

Line 209 - I am not sure which of the six scenarios is used as the "Second scenario" in your paper. Are these scenarios all differentiated by different levels of demand, and your paper is picking the one with the "Reference" demand scenario with 645 TWh of demand? This needs to be clearer.

Line 210-223 - This should be after Line 191 when nuclear plant closures are discussed. If this is presented at this juncture of the paper due to a systematic discussion of different energy sources, I'd say use different subheaders for the different energy sources, and add a sentence before that says that we are transitioning from the discussion of overarching scenarios to specific sources of generation.

Section 3.1 - I think when we are talking about energy mix, the scenario discussion should come AFTER the current generation sources are discussed, because otherwise it's a bit confusing when we are talking about nuclear power plants, then move to energy scenarios, and then move back to present-day energy sources.

In fact since 3.1 to 3.3 largely outlines the present-day state of the electricity /energy sector, I think any future projections should come after that, because that transitions better into the challenges. So a logical flow would be "Here's what's happening in France right now - here is what we expect to happen based on scenario designs - and here are the biggest challenges"

Line 353 - Line 363: Sure, demand can be adjusted to match production, but there's a human element to it that should be discussed as well. It's not like smart appliances with a two-way connection to the grid can change operating conditions at will without having some human intervention that allows it do so.

Hydrogen - Yeah sure, we can convert some of the excess generation into Hydrogen, but hydrogen in itself has limited uses (mostly industrial uses plus potentially heavy vehicles mobility and maybe some direct combustion usage for heating etc.). It's important to specify what's being done or should be done to increase the usage of hydrogen for the end-user, otherwise Hydrogen isn't really your top energy storage option.

EVs - Strictly speaking, PHEVs shouldn't be considered as a decarbonization option because people don't use them often enough on batteries. Literature supporting this exists from Plötz et al. (2020)

Pages 19 and 20 - Formatting is all over the place so the Energy Efficiency section is hard to read. I am not sure if it is on the authors or on the Editorial.

Line 761-766 - I am confused. How are these related to the two scenarios that were described in Section 3 ("negaWatt association scenario" and the "Energy Futures 2050 scenario with the Reference Demand case"). We were talking about two specific scenarios, and now all of a sudden we are talking about two families of scenarios. I am not even sure if those two scenarios are supposed to be examples of "Stability" and "Rupture" respectively or something completely different.

Line 769-770: Are these conclusions for BOTH scenario types or tailored for one? This needs to be made clear upfront.

Line 772 - is this still consistent with no nuclear power plants in 2045 in one of the scenarios and the PPE policy? Not very clear. I'm assuming no nuclear plants go beyond the average lifespan (about 50 years), so what additional measures are required that wouldn't be adopted already?

Line 776- Which particular fuel? Or is it a mix depending on specific circumstances (and define what those circumstances are)? Otherwise this is too vague.

Line 778 - I'm assuming flexibility refers to a way to store excess generation and a way to get energy back when renewables are not meeting the load requirements. This should be connected with the next set of findings more clearly.

Line 780 - I'm confused. The issue of carbon cycle didn't crop up in Section 4 once. Why is it being discussed for the first time here?

Line 782 - What is X?

Line 788-791: Really also applies for anything that uses batteries such as EVs. 

Line 792-796 - Also PHEVs shouldn't be encouraged

Line 815-818: Well if we are mentioning that here (and I think it makes sense because it's not tied to just energy storage), then the previous lines mentioning this are kind of redundant.

Line 823- 825: Not sure how COVID and high CO2 import reduction are related. This is not discussed in the main text at all. What exactly are "High CO2 emitting imports".

Line 833-837: This really needs to be more emphasized in the main text when all sorts of smart charging infrastructure are discussed, especially w.r.t. Sections 4.1 and Section 4.4

Author Response

Dear Reviewer

Thank you for giving us the opportunity to revise this manuscript. The authors appreciate the time and details provided by the editor and the reviewers. The insightful suggestions have certainly improved the manuscript. Below, you will find attached the detailed response to reviewers’ comments.

Reviewer 2 Report

  What are the benefits and what are the disadvantages of achieving carbon neutrality by 2050?
This must be stated in the abstract. The text format is very bad - various font sizes.
French energy is largely based on nuclear power plants.
Special attention must be paid to this topic.
The conclusion must also contain a clear overview of potential problems.
Also, a review of the situation in the surrounding countries.
That is, it must be placed in the context of France's developments in other countries.

Author Response

(The authors gave the same response as above.)

Round 2

Reviewer 1 Report

I think most of the major comments have been addressed. I don't see this paper as a major contribution to the conversation, but that doesn't stop me from recognizing this as a competently done research paper and recommend moving it to the next stage.

Author Response

  • The authors would like to thank the reviewer for this recommendation.

Reviewer 2 Report

This paper is significantly improved. Therefore, it can be accepted for publication.

My comments are minor - check font style through text as well as a Reference list.

Author Response

  • The authors would like to thank the reviewer for these remarks, the style and the reference list have been revised and updated in the manuscript.
